# Grief reaction and psychosocial impacts of child death and stillbirth on bereaved North Indian parents: A qualitative study

**Manoja Kumar Das**[1]*, **Narendra Kumar Arora**[1], **Harsha Gaikwad**[2], **Harish Chellani**[3], **Pradeep Debata**[3], **Reeta Rasaily**[4], **K. R. Meena**[3], **Gurkirat Kaur**[1], **Prikanksha Malik**[1], **Shipra Joshi**[1], **Mahisha Kumari**[1]

1 The INCLEN Trust International, New Delhi, India, 2 Department of Obstetrics and Gynaecology, Safdarjung Hospital and Vardhman Mahavir Medical College, New Delhi, India, 3 Department of Pediatrics, Safdarjung Hospital and Vardhman Mahavir Medical College, New Delhi, India, 4 Division of Division of Reproductive Biology Maternal and Child Health, Indian Council of Medical Research, New Delhi, India

* manoj@inclentrust.org

## Abstract

### Background

Grief following stillbirth and child death are one of the most traumatic experience for parents with psychosomatic, social and economic impacts. The grief profile, severity and its impacts in Indian context are not well documented. This study documented the grief and coping experiences of the Indian parents following stillbirth and child death.

### Methods

This exploratory qualitative study in Delhi (India) included in-depth interviews with parents (50 mothers and 49 fathers), who had stillbirth or child death, their family members (n = 41) and community representatives (n = 12). Eight focus group discussions were done with community members (n = 72). Inductive data analysis included thematic content analysis. Perinatal Grief Scale was used to document the mother's grief severity after 6–9 months of loss.

### Results

The four themes emerged were grief anticipation and expression, impact of the bereavement, coping mechanism, and sociocultural norms and practices. The parents suffered from disbelief, severe pain and helplessness. Mothers expressed severe grief openly and some fainted. Fathers also had severe grief, but didn't express openly. Some parents shared self-guilt and blamed the hospital/healthcare providers, themselves or family. Majority had no/positive change in couple relationship, but few faced marital disharmony. Majority experienced sleep, eating and psychological disturbances for several weeks. Mothers coped through engaging in household work, caring other child(ren) and spiritual activities. Fathers coped through avoiding discussion and work and professional engagement. Fathers resumed work after 5–20 days and mothers took 2–6 weeks to resume household chores.

**Data Availability Statement:** The datasets used and analyzed during the current study are uploaded as supplementary files.

**Funding:** This study is funded to The INCLEN Trust International by Bill and Melinda Gates Foundation (OPP1184205) through Indian Council of Medical Research (no 5/7/1504/2016- CH). The funders had no role in study planning, conduct, analysis and manuscript preparation.

**Competing interests:** The authors have declared that no competing interests exist.

**Abbreviations:** ASHA, Accredited Social Health Activist; FGD, Focus group discussion; IDI, In-depth interview; IQR, Interquartile range; MITS, Minimally invasive tissue sampling; PGS, Perinatal Grief Scale.

Unanticipated loss, limited family support and financial strain affected the severity and duration of grief. 57.5% of all mothers and 80% mothers with stillbirth had severe grief after 6–9 months.

## Conclusions

Stillbirth and child death have lasting psychosomatic, social and economic impacts on parents, which are usually ignored. Sociocultural and religion appropriate bereavement support for the parents are needed to reduce the impacts.

## Introduction

With 0.88 million under-five deaths, India topped the global list child mortality in 2018 [1]. About half of these children were neonates. An equivalent number of stillbirths also occur annually in India [2]. The under-five death is highest in central region followed by eastern region and the states including Uttar Pradesh, Bihar and Madhya Pradesh top the tally. At country level, the preterm birth complications (25.5%), intrapartum-related events (11.1%), sepsis (7.9%), congenital problems (6.0%), pneumonia (3.0%), and diarrhoea (0.4%) are the common causes of under-five deaths. The causes of death at regional and state level are similar [3]. Multiple pregnancy, maternal anemia, pregnancy induced hypertension, intrapartum and antepartum haemorrhage, obstructed labour, abnormal fetal presentation and congenital malformation are the common causes or risk factors for stillbirths in India [4].

Death of a child is one of the most severe, shattering and overwhelmingly painful event for parents. Bereavement is one of the most traumatic life events. Grief is a normal, but complex and multidimensional process of reactions to loss. Grief is dynamic, unique and usually individualised with variably affecting the physical, emotional, social, cognitive, and spiritual aspects [5]. The severity of grief after child death is affected by several factors, like age, gender, cause of death, unexpectedness, anticipatory grieving time [6–8]. Stillbirth is a unique devastating grief when a long anticipated birth is suddenly converted into loss and death. In neonatal deaths, a joy of birth soon gets crushed often by variable illness period and death. The grief following stillbirth and neonatal death, combined as perinatal grief is unique and may have long-lasting effects and even adverse outcomes [9, 10]. Despite no or brief physical and emotional relationship with their newborn, the perinatal grief doesn't significantly differ in intensity and manifestations from other bereavement scenarios [11]. In first-degree relative bereavement situations, the grief symptoms intensity usually decline over the 6–12 months and sometimes may take 24 months [12–14]. The perinatal losses and child deaths have a considerable psychosomatic impact on parents and the family, and may cause post-traumatic stress, depression, anxiety, eating and sleeping disorders, chronic diseases aggravation and lower quality of life [15–17]. Although majority of the bereaved parents cope with the grief, some have complicated grief, which was observed to be higher with child death [18].

Over last few decades there is increased recognition of the perinatal, infant and child death related grief and associated psychosocial impacts in developed countries. A gradual and widespread sociocultural change in discussion and acceptance of the perinatal losses for parents and families has been observed. There has been development of targeted bereavement and grief counselling programs, support groups, and social media platforms to address the issues and enable better coping by parents [19]. But such change is not visible in developing

countries, may be due to limited documentation and understanding of the factors and socio-cultural contexts.

Despite high burden of child deaths and stillbirths in India and other developing countries, the grief characteristics and phenomenology for parents, families and societies remain neglected [20]. The scant reports from Central and South India found that the parents suffered from serious form of grief, guilt and mental health challenges following perinatal loss [21, 22]. There is need to understand the grief characteristics among the parents and family level changes after child death and stillbirths from different parts of India and also the various coping strategies adopted by them within their social environment. This study explored the social, emotional and psychological impact of child death and stillbirths on parents and their families in the north Indian context.

## Methods

### Study setting

This study was conducted in Delhi urban area anchored to a tertiary care government hospital, where a pilot project for minimally invasive tissue sampling (MITS) to identify the causes of child and neonatal deaths and stillbirths was undertaken. Prior to initiation of the MITS for deceased children and stillbirths, a formative research was conducted to understand the responses of the parents and community to child death and stillbirth including the grief, coping mechanisms and support systems experienced by them. This formative research was conducted during September 2018 to April 2019. The anchoring hospital provides free-of cost treatment and adopts no-refusal policy. The hospital is usually accessed by all types of patients, but primarily from low and middle socioeconomic classes from Delhi and adjoining northern states. It conducts about 27000 deliveries and admits about 16000 children annually, and thus is always overcrowded.

### Study design

This study adopted exploratory qualitative research design. In-depth interviews (IDIs) and focus group discussions (FGDs) with various participants were conducted at their respective locations.

### Study participants

Two categories of participants were included: (i) parents who had child or neonate death and stillbirths at the hospital and their family members; (ii) community representatives including influential community leaders, community health functionaries, and religious leaders. The parents of the children who died and stillbirths that occurred at the hospital were identified from hospital registers/records and contacted over phone and/or visited at home 6–8 weeks after the unfortunate event. The parents from different religions and localities of Delhi with both parents available were identified and approached for consent. The parents residing outside Delhi were excluded. In total during the reference period, there were 45 child deaths, 52 neonatal deaths and 60 stillbirths. Out of these 69 parents were not traceable due to incomplete address or no contact number. We were able to contact 30 eligible parents with child death, 28 parents with neonatal death and 40 parents with stillbirth. Out of these approached, 13 parents with child death, 12 parents with neonatal death and 22 parents with stillbirth consented for IDI. The community representatives from localities were purposively identified for IDIs and FGDs. For the FGD we considered four categories of participants; mothers, fathers, elder family members including grandfathers and grandmothers with child under-five years, but not

had child death or stillbirth. The detailed study protocol has been published earlier and some findings of the study regarding MITS acceptance have been published [23, 24].

## Data collection

Data were collected through semi-structured IDIs and FGDs guides developed based on the objectives and literature (S1 File). The IDI guides for parents had open-ended questions that explored the issues including grief experience, interpersonal and family level challenges, coping mechanisms, support systems and return to work/normalcy. The IDIs for community and religious members and FGDs explored their broad perspectives about the grief and coping by parents in the society/community. Additional probing were done, as needed based on the responses. The IDI and FGDs guides were pilot tested and finalised based on the findings. Two pairs of female researchers (GK, PM, SJ, MK, females) trained in qualitative research and interviews conducted the IDIs at homes or places convenient for participants. The interviews with mother and father of the deceased child or stillborn were conducted separately at their households. The IDIs with parents were continued till no significant new issues or themes emerged indicating data saturation and the participants with different sociocultural background were included. After data saturation 2–3 additional IDIs were conducted. The FGDs were conducted at convenient and neutral venues in the community with 8–11 participants from similar category. It was facilitated by the investigator (MKD, male) experienced in conducting IDI, FGD and qualitative research with support from the four female researchers. During the IDIs/FGDs, other person's presence was avoided as far as possible. The teams conducted IDIs/FGDs in local language (Hindi) and were audio-recorded with permission. The team also took detailed field notes capturing the verbal and non-verbal expressions. For documenting the quality of FGDs and dominance of any participant, sociograms were drawn (S3 File). The average time taken for the IDIs and FGDs were 45 minutes and 60 minutes, respectively.

We contacted the mothers after 6–9 months of child death or stillbirth to capture the grief status using the Perinatal Grief Scale (PGS). PGS is a 33-item questionnaire with five point Likert scale ranging (1) strongly agree to (5) strongly disagree [25, 26]. The scores for two items are reversed and totalled to find the total PGS score (range 33–165) and higher scores represent more severe grief. A total PGS score above 91 indicates high degree of grief [26]. The Hindi translated version of PGS available at https://judithlasker.com/perinatal-grief-scale/ was used with due permission, which have been used in other studied in India [27–29]. The scale includes three sub-scales on active grief, difficulty coping and despair with 11 items each. The questions were readout to the participants by the research staff and the responses were recorded. Any response needed clarification or further exploration were clarified or further probed. All the interviews and individual data collection were done at the households of the participants and FGDs were conducted at community level. No payment was made to any participant.

## Data handling and analysis

The team transcribed the audio recordings and field notes in local language, which were translation into English (S2 File). The transcriptions correctness and quality were checked by different members with the audio-records and transcriptions. The data was entered using INCLEN Qualitative Data Analysis Software, which allows data entry, organization and retrieval for analysis in Indian languages and English. The entered data (transcripts) were checked for correctness (in reference to the transcripts) and completeness by another member. The data were saved into the server and backed up regularly. For analysis we used constructivist grounded

theory as the methodological strategy [30]. The grounded theory explores the issues inductively from the data collected, not testing pre-assumed hypothesis. Two researchers independently read through the IDI and FGD transcripts several times and listed the possible codes from the lines, segments and incidents. The IDIs and FGDs were separately coded. The codes were examined for identification of the similarities and differences and grouped under themes. The codes and themes were reviewed to identify the linkages (axial coding) and grouped into fewer categories (selective coding) and assembled under key themes. A reiterative process for the transcript reading, coding and thematic summarization was followed till the investigators agreed on the final framework. The themes across the participant categories and methods were triangulated for consistencies and differences. The findings were expressed semi-quantitatively using qualifiers: very few (<10%), some (10–24%), about half (25–49%), majority (50–75%), most (76–89%) and almost all (>90%). The PGS scores (total: 33–165 and sub-scales: 11–55) were calculated and the sum score was expressed as median with interquartile range (IQR) for all mothers and sub-groups (Cronbach's alpha = 0.91) (S4 File). The higher scores indicate more severe grief.

### Ethical considerations

The participants were recruited after obtaining written informed consent including permission for audio recording and use of anonymized quotes. Confidentiality and anonymity of participants were assured. Ethical approval for the study was obtained from all the participating institute ethics committees (The INCLEN Trust International, Ref: IIEC 51 and V.M.M.C. and Safdarjung Hospital, Ref: IEC/SJH/VMMC/Project/August-2017/1000).

## Results

This study included 25 families with deceased children (neonates, n = 12 and >1month, n = 13, participants, n = 49 parents and 21 family members) and 22 stillbirths (n = 44 parents and 20 family members). The community participants included community members, community health functionaries and religious leaders (n = 12, 4 from each category). The eight FGDs included 72 community participants (8–10 per FGD). The demographic information about the IDI and FGD participants are given in Table 1. The causes of death for the 25 child deaths according to the hospital records are summarised as S5 File.

Inductive analysis of the data resulted four broad themes and subthemes: (1) grief anticipation and expression: (1a) anticipation of death or stillbirth, (1b) immediate emotional outpouring, (1c) physical reactions and (1d) grief and reactions; (2) impact of the bereavement: (2a) guilt, remorse and blames, (2b) changes in relationship, (2c) changes in lives and livelihood; (3) coping mechanism and (4) sociocultural norms and practices. The coding tree for the themes and categories are given as S6 File. The findings under the themes and sub-themes along with the illustrative statements from participants are presented below.

### Grief anticipation and expression

**Anticipation of death or stillbirth.** Majority of the parents of children were communicated about the serious condition of the child at admission or during the hospitalisation. Most of the parents of newborns were informed about the condition and the risks soon after delivery or at the time of admission. Some of the mothers of stillborn were apprehensive due to decreased or absent foetal movements. Few mothers were told about the intrauterine death on ultrasound prior to the delivery.

**Table 1. The sociodemographic characteristics of the study participants.**

| Sl no | Parameters | Results | |
|---|---|---|---|
| | | **Value** | **IQR** |
| *1* | *Parents participated in IDIs* | | |
| 1.1 | Age (in years), median (IQR) | | |
| | • Mother (n = 47) | 25 | (22–30) |
| | • Father (n = 46) [a] | 29.5 | (27–34) |
| | • Other family members (n = 41) | 56 | (48–65) |
| 1.2 | Religion (n = 47) | | |
| | • Hindu, n (%) | 27 (57.4) | |
| | • Muslim, n (%) | 18 (38.3) | |
| | • Christian, n (%) | 2 (4.2) | |
| 1.3 | Literacy of mother (n = 47) | | |
| | • <5th standard, n (%) | 11 (23.4) | |
| | • 6th -10th standard, n (%) | 18 (38.3) | |
| | • >10th standard, n (%) | 18 (38.3) | |
| 1.4 | Literacy of father (n = 46) | | |
| | • <5th standard, n (%) | 6 (13) | |
| | • 6th -10th standard, n (%) | 20 (43.5) | |
| | • >10th standard, n (%) | 20 (43.5) | |
| 1.5 | Mother's occupation (n = 47) | | |
| | • Housewife, n (%) | 46 (98) | |
| | • Working (skilled worker), n (%) | 1 (2) | |
| 1.6 | Father's occupation (n = 46) | | |
| | • Skilled worker, n (%) | 21 (45.6) | |
| | • Self-employed or business, n (%) | 21 (45.6) | |
| | • Daily wage laborer, n (%) | 4 (8.7) | |
| *2* | *Family members participated in FGDs* | | |
| 2.1 | Age in years, median (IQR) (n = 72) | | |
| | • Fathers (of children aged < 5 years) (2 FGDs, n = 18) [c] | 28 | (25–30) |
| | • Mothers (of children aged < 5 years) (2 FGDs, n = 19) [c] | 26 | (25–30) |
| | • Grandfathers (2 FGDs, n = 17) [c] | 58 | (53–62) |
| | • Grandmothers (2 FGDs, n = 18) [c] | 58 | (52.5–61) |
| *3* | *Community representatives participated in IDIs* | | |
| 3.1 | Age in years, mean (range) (n = 12) | | |
| | • Influential community leaders (n = 4) | 58 | (54–62) |
| | • Community functionaries for mother and child (n = 4) [b] | 47 | (44–50) |
| | • Religious leaders (n = 4) | 57 | (52–62) |

Note:

[a] Father of one deceased child died some months ago;

[b] Includes Anganwadi workers (community maternal and child nutrition workers) and ASHAs (Accredited Social Health Activists, community maternal and child health workers) engaged in public health/nutrition services;

[c] The participants for FGD didn't belong to the families who had child death or stillbirths included in the study.

Abbreviations: ASHA, Accredited Social Health Activist; FGD, focus group discussions; IDI, in-depth interview

> "It was Sunday and I was on night duty. She felt less movement and pain during the day and night. I came on Monday morning. She had some relief in pain by then. In the evening when it worsened, we went to hospital. After checking they told that there was no heartbeat and referred to another hospital. They did ultrasound and told that the baby was dead."

*(Father of stillborn)*

*"It had been three days since the baby was dead. They were not doing the delivery. I was requesting the doctors for delivery, because I was feeling pain and why wait further."*

*(Mother of stillborn)*

Majority of the children and newborns who were on respiratory support, parents noted the sudden worsening and called the doctors/nurses. Some of the mothers were worried with the delivery process: longer duration, assistance or difficult delivery.

*The doctors in other hospital (where she delivered) forcibly did normal delivery. That's why the baby had problem and didn't cry.*

*(Mother of deceased neonate)*

*"I observed that the hands and feet became cold, the chest was warm, but child was not breathing. I called the doctor. They came tried and then declared my child to be dead."*

*(Mother of deceased child)*

*Immediate emotional outpouring*: Usually the news of death or stillbirth were delivered to the father or other family member present, not to the mothers. Prior to the death declaration, the doctors/nurses asked the family to move the mother away from the bedside/ward. For majority of stillbirths, the doctors/nurses told the mother that the baby was taken to the nursery. According to the mothers, sharing the same ward or room with the mothers with live baby increased their suffering.

*"While my husband cried in hospital after listening about death of baby but still controlled himself to support me."*

*(Mother of deceased newborn)*

*"The guards were very strict. They did not allow any male person inside the ward. There were mothers who had live baby and also who had lost their baby. It was difficult time."*

*(Mother of stillborn)*

**Physical reactions.** Majority of the parents were shocked to hear the news. About half of the mothers fainted in the hospital itself on hearing the news. The mothers of five children/neonates and two stillborn were not told about the death/stillbirth at the hospital and taken to home, where they were informed by their husbands or family members. All the mothers cried loudly on hearing the news. The mothers present at the hospital saw their child/baby before covering and handover. Most fathers expressed that they were shocked, extremely sad and disturbed, but could not express openly. Some of the fathers cried on hearing the news at the hospital, but soon controlled themselves perceiving the possible impact on the mother.

*"My husband is sad. But he doesn't share his sadness, thinking by doing this everyone will remain sad"*

*(Mother of deceased child)*

*"Child's father cried and begged to doctor for saving his child. He still visualizes the child, runs outside the house saying my child is calling me."*

*(Grandmother of deceased child)*

**Grief and reactions.** The parents remembered that they experienced mixed feeling including disbelief, unimaginable pain, helplessness and vacuum with loss of their child. The parents who had stillbirth expressed that their long awaited expectations of pregnancy suddenly transformed into something unimaginable and many of them were unsure how to react. Several of them had anger and dissatisfaction against the hospitals or care providers as the baby died despite regular check-ups and tests during pregnancy. Majority of parents who had stillbirth had apprehension about the next pregnancy and were unsure where to go and what more to be done.

*"One and half month has passed since this event, we are not able to get how all this happened when everything was going fine, then how did this happen? We are in suspense."*

*(Father of stillborn)*

*"I still visualize the child and feel as if the child is drinking breast milk. Milk flows and wets the bed."*

*(Mother of deceased child)*

## Impact of the bereavement

**Guilt, remorse and blames.** Some parents had regrets about some decisions and actions they had made like not seeking care early, not being present, taking or not taking to a hospital. Few mothers mentioned of being blamed by their husband and in-laws for the loss. Few mothers blamed their in-laws for not appropriate care during pregnancy.

*"I blame myself. I should have taken child to hospital in night itself, then he would have been alive, there was delay at home only"*

*(Father of deceased neonate)*

**Changes in relationship.** Almost all parents indicated that the death/stillbirth had affected their relationships. While majority of the couples indicated positive change in their relationship like the husbands had become supportive. Two mothers who had stillbirth reported neglect by their husbands after the event and one of them was nearing separation. Most of the mothers were supported by their husbands and also the immediate/extended family members for variable periods, ranging from two days to few weeks. These family members included in-laws (mother-in-law and sister-in-law) and the maternal family members (mother's sister and mother). Two couples reported no support from the other family members.

*"After child's death, her in-laws fight with my daughter; her husband and her father-in-law were ready to leave my daughter. Her husband was ready to give her divorce. This all happened after the child death. They blamed my daughter that why you took child to this hospital".*

*(Maternal grandmother of deceased child)*

*"My husband remains tense. Even he became jobless that time. He even says due to your negligence all these have happened, you didn't take him on time to hospital. Now we fight more, and also we are not able to take-care of our other child."*

*(Mother of deceased neonate)*

**Changes in lives and livelihood.** Some of the mothers had continued illness, psychological problem and poor eating for several weeks after the event. Half of the mothers reported weeping in isolation and remembering the baby by seeing the photos. Majority of the mothers and fathers continued visualizing their child to be either present in house or playing outside or feeling as if the baby is breastfeeding. Some parents reported that they couldn't eat properly, had sleep disturbance and high blood pressure after the event for some days. Long absence from work or loan for treatment led to financial problems in few families, which add to grief. Few families took loan to meet the expenses related to the event.

*"I am not able to sleep properly at night. My mind constantly think about the child. Fourteen years have passed to marriage, if we had one child, I wouldn't worry."*

*(Father of stillborn)*

*"My husband did not go to work for 1 month. He was with me for a month. He left the job after that. Now he goes to work when some work comes."*

*(Mother of stillborn)*

## Coping mechanism

Most of the parents had accepted the loss over time and tried to restore normalcy. As majority of the participants were from low socioeconomic class, the fathers returned to work after 5–20 days. One father had to leave the job and three had changed the job due to longer absence and inability to concentrate on work. Majority of the mothers resumed household work 2–4 weeks after the event. But, four mothers had continued problem and couldn't resume work until 6 weeks. Majority of the mothers tried to keep themselves busy with the care of other child/children and household work. They also increased their engagement in spiritual and religious activities (prayer/namaz/rituals). Few mothers preferred isolation and remembered their child. Few mothers were pregnant at the last interaction. The fathers mentioned keeping themselves busy in work to cope with the loss. Some of them mentioned that they preferred not to discuss about the loss before their wives and family. Few fathers mentioned of remembering the child/baby in isolation.

*"My condition was not good and felt as if I am going to die after listening about child death, I divert my mind by reading namaz (prayer). I got support from my sister-in-law and other children. I started doing household work after 40 days."*

*(Mother of deceased child)*

*"How am I coping with this, only I know. My first child is no more, how I will cope with this. Sometimes I am sitting in sadness and thinking that maybe I also had baby."*

*(Mother of deceased newborn)*

*"I am not able to cope, I still remember my baby and I am not able to concentrate, I have headache and I am not doing household work after the event"*

*(Mother of stillborn)*

## Sociocultural norms and practices

The community members and religious leaders mentioned that the rituals for the stillbirths and child death were different from adults. Although burial practice was followed for all across religions, the rituals had some variation according to the age. The women were not allowed to the burial ground. The families follow rituals according to their religion and beliefs including mourning and prayers. Additionally some families do activities like feeding and donation to poor. According to them sometimes the families blame the woman or witchcraft for the death/loss.

*"Yes, many families usually blame the woman for the death. Some people consider death of a child inside womb because of some witchcraft (kala jadu, jadu tona)."*

*(Religious leader, Hindu)*

*"The sorrow is sorrow and it remains somewhere inside. But person need to live and carry on with the family and work. Everyone experiences sorrow sometime and happiness sometime. Slowly they overcome from the sorrow engaging themselves in family and work. They even seek help from almighty (God) and do regular namaz."*

*(Religious leader, Muslim)*

## Grief status after 6–9 months

We could collect the grief scores using PGS from 40 mothers (13 with child death, 12 with neonatal death and 15 with stillbirths) after 6–9 months of the loss (Table 2). The median perinatal grief score was 107 for mothers with stillbirth (above the cut-off score 91), indicating high level of persisting grief compared to the mothers with child/neonatal deaths (median PGS scores 86) (p = 0.02). The scores for the three PGS subscales were consistently higher for mothers with stillbirth compared to the mothers with child/neonatal death and was significant for despair subscale (p = 0.02; active grief subscale, p = 0.11; difficulty coping subscale, p = 0.17). Out of the mothers, 57.5% had severe grief (total PGS score >91). Compared to the mothers of deceased children/neonates, higher proportion (80%) of the mothers with stillbirth had severe

**Table 2. Mother's grief status after 6–9 months of the event by Perinatal Grief Scale (PGS).**

| Mother' category | Active grief sub-scale Median (IQR) | Difficulty coping sub-scale Median (IQR) | Despair sub-scale Median (IQR) | Total PGS score Median (IQR) | Severe grief n (%) |
|---|---|---|---|---|---|
| Deceased child (n = 13) | 36 (29–39) | 26 (25–30) | 27 (22–32) | 86 (79–102) | 6 (42.8) |
| Deceased neonate (n = 12) | 29 (27–29) | 28 (25–29) | 26 (24–28) | 86 (73–97) | 5 (45.4) |
| Stillbirth (n = 15) | 41 (32–42) | 32 (25–33) | 35 (27–43) | 107 (97–117) | 12 (80) |
| Pooled (n = 40) | 36 (28–42) | 29 (25–32) | 28 (23–34) | 97 (78–107) | 23 (57.5) |

Note: Severe grief: Total PGS score >91; IQR: Interquartile range

grief. It was observed that majority of the mothers had higher scores in the active grief sub-scale.

## Discussion

This study is one of the only few exploring the experience from bereaved parents and families in India. We observed that the parents suffered from grief, disbelief, severe pain and helplessness on hearing the death/stillbirth. The doctors and family usually preferred not to declare the death/stillbirth to the mother directly. All mothers expressed severe immediate grief reaction including fainting for some. Fathers also suffered from severe grief, but didn't express openly and tried to support their wives and family. Some parents shared about the guilt and blames targeted at the hospital/healthcare providers, themselves or the mother by the family. Although majority of the parents had either positive or no change in the couple relationship, few indicated marital disharmony of various degrees. Majority of the parents had some form of sleep, eating and psychological disturbances, which continued for several weeks. Most of the parents were supported by the immediate and extended family. The mothers tried coping with the bereavement and grief through engaging in household work, care of other children and engaging in spiritual activities. The fathers tried coping through work and professional engagement and avoided discussion about the event with wives/family. While the fathers usually returned to work after 5–20 days, the mothers took longer (2–6 weeks) to return to the household chores. The factors like unanticipated loss, limited family support and financial strain affected the severity and duration of grief. After 6–9 months of the loss, persistence of severe grief was observed among 57.5% of mothers. A higher proportion (80%) of mothers with stillbirth had severe grief.

Children are symbolic representation of parents' reproductive capacity, future hope and social capital. With child death or stillbirth, the parents perceive it as loss of a part of self, end of dream and loss of competence and power. The grief reactions reflect their threatened position as the protector and caregiver of the child, which are rationalised through the guilt, self-blame, anger and other behavioural manifestations [31].

The limited information on bereavement and grief from India primarily focus on the still-births, from Chhattisgarh (Central India) and Chennai (South India) [21, 27, 22]. The study from central India including tribal women observed that the women experienced extreme sadness, unhappiness, shame, dishonour, worthlessness and social blame following stillbirth. They faced psychosomatic problems including lack of concentration, sleep disturbances, and eating problems. They perceived loss of the social status, dignity and identify to various degree. All of them had desire of being pregnant again, but with apprehension. The perinatal loss and grief was not adequately recognised as an issue by the society, family and healthcare providers [21]. The study from urban South India documented that following stillbirth the women experienced severe grief, guilt, and remorse. They expressed dissatisfaction, frustration and anger regarding the healthcare experience. The factors that aggravated the grief were insensitive family, friends and neighbours, strained marital relationship, social stressors, financial strain and unresponsive healthcare. Several positive and negative coping mechanisms reported were isolation, support from friends and family and engaging in care of other children. Most of the respondents expressed need for better response from the healthcare providers during the post-delivery phase [22]. The findings from these two qualitative studies are comparable to the findings from our study.

Another study in Chhattisgarh documented average PGS score of 110.04 among the women with stillbirth 1–2 years ago. The active grief sub-scale scores were higher (mean 42.09) than the other sub-scales (33.87–34.09). This study also piloted socio-culturally adapted

family level intervention with improvements in the perinatal grief and mental health symptoms [29]. The PGS scores among the mothers with stillbirth (median 107, IQR 97–117) and proportion (80%) of mothers with severe grief in our study were comparable to the findings from Chhattisgarh. This probably indicate the possible severe and persisting grief following stillbirth compared to the child/neonatal deaths.

A study from north India reported severe grief reaction (75.3%), remorse (72.0%), depression (48.6%), anxiety (51.6%) and self-blame (36.9%) among mothers following stillbirth. The mothers suffered from stigmatization (76.2%), rejection by family (24.8%), spousal abuse (19.2%) and disturbed relationship with extended family (10.8%). The noteworthy changes in spouse and family relationship reflected the deep rooted societal norm and family dynamics. The grief reactions and family level changes reported were comparable to the observations in our study [32].

A study among Spanish parents with stillbirth reported grief reactions including shock, disbelief, denial, despair, hopelessness and anger following the loss. A delay in providing information, inadequate explanation about the cause and sharing room with mothers who had livebirth aggravated the suffering. Some parents wanted quick removal of the dead baby. Mothers felt lonely and lost even if surrounded by many people and wanted the company of their husbands. The support from healthcare providers in assembling mementos of the deceased baby (foot/hand prints, umbilical cord clamp, and clothes) were appreciated. The refusals for performing rituals by hospital staffs increased their sufferings [33]. In our study also few mothers who knew about intrauterine death of the baby, wanted quick delivery after declaration, even if needed surgery. The mother's suffering increased with sharing the ward/room with mothers with livebirths. In our study also mothers wanted presence of their husbands during post-delivery hospital stay.

Mothers in Ghana indicated that they saw their babies after death but wanted to hold and spend more time due to the rituals. No memorial services were held for the deceased infants. The mothers were supported by family, but were explicitly discouraged from discussing about the loss. They coped with grief through acceptance of the loss, isolation, focusing on care of other children and spiritual activities. Some mothers found avoidance of discussion or thinking about the child was painful. There was no change in the relationship with husbands after the loss [34]. There were similarities in the experiences and coping strategies of the mothers in our study and from Ghana. The cultural silence regarding the perinatal and infant loss were also similar.

Parents from Ireland with perinatal loss experienced emotional and stress conflict with the declaration. They perceived the baby to have a personhood and unique identity that influenced their lives with protective instincts. They had regrets and guilt about some care-related actions and decisions they had made. While most of the parents had some negative impact on relationship with their partner and difficulty in communicating their grief feelings [35]. In another study parents from United Kingdom wished to hold their babies longer and collect meaningful physical mementos. They felt that caring and humane professional support from hospital staff would assist in parental recovery [36]. In Netherland, severity and persistence of grief was associated with female gender, death of a child, and lower educational status. The grief persisted in 30% of the individuals even after six years of the loss. The persons with higher grief score at baseline had higher risk of persistence [37]. This suggested the need for support for the individuals with severe grief manifestations soon after the loss.

Among the British parents with child death, grief was observed variably until five years after the event. The prolonged and higher grief were associated with low levels of optimism, cognitive restructuring, high levels of avoidance, depression, pessimism, self-blame and alcohol/substance use. Out of these factors avoidance, depression and low levels of cognitive

restructuring were leading influencing factors for the severe and prolonged grief. Financial difficulties due to bereavement was associated with depression symptoms [38]. Among South African mothers who had infant death the grief severity declined over 6–30 months. Over time, the prevalence of avoidance, meaninglessness feelings, shock, and numbness declined, but diminished sense of self, anger, bitterness, lack of trust on others persisted [39].

Mothers suffer from higher degree of grief, which also last longer than the fathers. The studies in European and North American context observed greater and longer grief among mothers compared to fathers up to four years [40–42]. By 18 months while mothers had better control over grief, but fathers were still angry and openly expressed their grief [42]. Although the fathers react differently, they also suffer from strong emotional reactions, somatic symptoms, and social interaction difficulties [43, 44]. Decrease in grief was observed over 3–13 months for mothers and over 3–6 months for fathers after neonatal/child death. The fathers perceived their grief to be equally severe as the mothers after the child death, but they expressed differently [45]. In our study, we also observed that the fathers experienced severe grief following the loss, but expressed differently by putting a strong face to support their wives and avoiding discussion. The sociocultural context, professional and educational status influence the varied grief reactions by fathers.

In the traditional and paternalistic societies, the perinatal grief and post-child death grief are often ignored and not discussed. This may be due to long history of the high child mortality and stillbirth burden in the country, which has prepared the society to accept these losses as norm. The Indian society primarily functions as relationship-centred society and encourages expression of the feelings (both positive and negative), crying, anger and dependence on others. It also has beliefs and valued centred around the spiritualism, not materials like the western societies [46]. While the grief reactions have been better studied over last few decades in the western contexts and several grief support mechanisms and protocols have emerged, it is still in nascent phase in developing countries and societies like India.

The clinical and public health strategies have primarily focused on the biomedical causes of mortalities. Healthcare system and providers in developing countries are often either dismissive or ignorant of the potential impacts of the perinatal loss or child deaths. This may be reflection of the societal norm of recognition and giving identity to the stillborn and young infants. Simultaneously, the psychosomatic, familial/social and economic consequences experienced by bereaved parents and the determinants have been relatively ignored in the Indian context. The documentation of the burden, experiences of parents/families and the determinants would improve the awareness and evidence base for development of sociocultural and religion appropriate protocols including communication packages and training of the healthcare providers. Considering the potential impact on the parents and families, there is need for provision of bereavement and grief support at the hospital levels and active involvement of the nurses and doctors in the process. The grief support and counselling should be extended at the community level through the community health and nutrition functionaries along with future pregnancy planning and birth spacing, as needed. There is need to encourage establishment of community level support systems, bereavement helplines and professional groups for providing appropriate guidance and support to the parents using sociocultural and religion compatible materials, as available in United Kingdom, United States, Canada, Australia and some European countries.

The study had some limitations. The findings from the participants of this study may be context specific and hence may not be generalizable to other settings. We captured the grief scores for the mothers once. The grief scores for fathers was not collected.

## Conclusions

In Indian context, the stillbirth, infant and child death are primarily considered as a clinical problem and ignores the significant and long lasting psychosomatic, social and economic impacts on the parents and family. While the mothers experience severe grief and express openly, the fathers although suffer equally, avoid open expression. Even after six months of the loss, more than half of the mothers had severe grief. The mothers had greater grief severity after stillbirth compared to neonatal/child death. There is need for further documentation of the grief characteristics with stillbirth and child deaths, impact at individual and family level, influencing factors and dynamics with time from diverse sociocultural contexts in India. Also there is need for recognising the need of care, counselling and support for the parents after stillbirth and child death at societal, institutional and programmatic levels.

## Supporting information

**S1 File. The study tools and guides used for data collection (in-depth interview guides, focus group discussion guides and perinatal grief scale).**
(PDF)

**S2 File. Transcription of the in-depth interviews and focus group discussions with the participants.**
(PDF)

**S3 File. The sociogram for the focus group discussions conducted.**
(PDF)

**S4 File. The perinatal grief scale score data for the participants.**
(XLSX)

**S5 File. The causes of death for the children and neonates (whose parents participated in the study).**
(PDF)

**S6 File. The coding tree derived by inductive analysis of the IDIs and FGDs.**
(PDF)

**S1 Checklist.**
(DOCX)

## Acknowledgments

We acknowledge the participation of the parents and community members for their contribution. We appreciate the participation and support from paediatricians, obstetricians, residents, nurses and record section officials of VMMC and Safdarjung Hospital, New Delhi in conduct of this study. We highly value the guidance from the Technical Advisory Group members, Dr Siddarth Ramji, Maulana Azad Medical College, New Delhi; Dr Gagandeep Kang, Translational Health Science and Technology Institute, Faridabad, Haryana; Dr Sunita Saxena, National Institute of Pathology, New Delhi; and Dr Yogesh Jain, Jan Swasthya Sahyog, Bilaspur, Chattisgarh. We acknowledge the cooperation from the co-investigators: Dr Usha Agrawal and Dr Fauzia Siraj, National Institute of Pathology (Indian Council of Medical research), New Delhi; Dr Pratima Mittal, Dr Rajni Gaind, Dr K.C. Agarwal, Dr Archana Kashyap and Dr Manisha, Safdarjung Hospital and Vardhman Mahavir Medical College, New Delhi. We also acknowledge the support from other INCLEN team members including Deepak Singh, Vinod

Kumar, Chandan Singh, Amit Kumar, Bablu and Rajender. We appreciate the technical assistance received from CHAMPS project team members.

## Author Contributions

**Conceptualization:** Manoja Kumar Das, Narendra Kumar Arora, Reeta Rasaily.

**Data curation:** Manoja Kumar Das, Harsha Gaikwad, Harish Chellani, Pradeep Debata, Reeta Rasaily, K. R. Meena, Gurkirat Kaur, Prikanksha Malik, Shipra Joshi, Mahisha Kumari.

**Formal analysis:** Manoja Kumar Das, Gurkirat Kaur, Prikanksha Malik, Shipra Joshi, Mahisha Kumari.

**Funding acquisition:** Manoja Kumar Das, Reeta Rasaily.

**Investigation:** Gurkirat Kaur, Prikanksha Malik, Shipra Joshi, Mahisha Kumari.

**Methodology:** Manoja Kumar Das.

**Project administration:** Manoja Kumar Das, Narendra Kumar Arora.

**Resources:** Manoja Kumar Das, Narendra Kumar Arora, Reeta Rasaily.

**Supervision:** Manoja Kumar Das, Narendra Kumar Arora, Harsha Gaikwad, Harish Chellani, Pradeep Debata, Reeta Rasaily, K. R. Meena.

**Validation:** Manoja Kumar Das, Gurkirat Kaur.

**Writing – original draft:** Manoja Kumar Das, Gurkirat Kaur.

**Writing – review & editing:** Manoja Kumar Das, Narendra Kumar Arora, Harsha Gaikwad, Harish Chellani, Pradeep Debata, Reeta Rasaily, K. R. Meena, Gurkirat Kaur, Prikanksha Malik, Shipra Joshi, Mahisha Kumari.

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
