## [Decision Letter · Decision Letter 0]

16 Oct 2020

PONE-D-20-30229

Grief reaction and psychosocial impacts of child death and stillbirth on bereaved North Indian parents

PLOS ONE

Dear Dr. Das,

Thank you for submitting your manuscript to PLOS ONE. After careful consideration, we feel that it has merit but does not fully meet PLOS ONE’s publication criteria as it currently stands. Therefore, we invite you to submit a revised version of the manuscript that addresses the points raised during the review process.

Please go through the comments of the reviewer and revise your manuscript. 

We look forward to receiving your revised manuscript.

Kind regards,

Vijayaprasad Gopichandran

Academic Editor

PLOS ONE

Journal Requirements:

"This study is funded to The INCLEN Trust International by Bill and Melinda Gates

Foundation (OPP1184205) through Indian Council of Medical Research (no 5/7/1504/2016-

CH). The funders had no role in study planning, conduct, analysis and manuscript preparation."

"None"

"None"

Reviewers' comments:

Reviewer's Responses to Questions

**Comments to the Author**

1. Is the manuscript technically sound, and do the data support the conclusions?

Reviewer #1: Yes

2. Has the statistical analysis been performed appropriately and rigorously? 

Reviewer #1: N/A

3. Have the authors made all data underlying the findings in their manuscript fully available?

Reviewer #1: Yes

4. Is the manuscript presented in an intelligible fashion and written in standard English?

Reviewer #1: Yes

5. Review Comments to the Author

Reviewer #1: This is a well-written qualitative paper which aims to study the social, emotional, and psychological impact of child death and stillbirths on parents and their families along with the coping strategies in North Indian context. I appreciate the authors for their commendable work as this is particularly important and less investigated topic of women and child health.

Comments

Title

I would suggest the authors to add “a qualitative study” at the end of the title as the readers will know the type of study was conducted for this research question.

Abstract

The abstract summarizes the rationale for the study and the methods used for this study, as well as the key findings.

Introduction

The introduction includes important information about the global burden of neonatal and infant mortality, the bereavement and grief caused to the mother and family after infant death and shows the paucity of information on the social and psychological impact after child death in North India.

Line 54- I would suggest the authors to write few more lines on the epidemiology of child death in India.

At the end of line 54, authors should discuss about the consequences of child death in the family and then the statements on bereavement and grief should be discussed, as it would give the readers a sense of continuity. Right now, it looks separate because the burden was discussed in the initial few lines of Introduction followed by definition of bereavement and grief.

Line 85- Authors had mentioned “causes of a large proportion of child, infant deaths and stillbirths remain unknown”. This statement should be removed as the leading causes of infant and neonatal deaths are well-documented. I would suggest the authors to write a few lines on the common causes of neonatal and infant deaths in India along with the description of epidemiology of child deaths in the 1st paragraph of Introduction.

Methods

The methods section provide details on how the participants were chosen, how the interview was conducted, and all the required information needed to understand the whole process of the study.

Line 114- I would suggest the authors to provide some information on how many refused to participate in the study among those who were contacted to find whether there is any systematic difference in the socio-demographic characteristics between the participants and non-participants.

Line 143- There is no information on the internal validity (Cronbach’s alpha) of the scale used in this study.

I suggest the authors to comment on the fidelity of the transcription. Whether it was read out to participants after each session to check for correctness or any other method was adopted.

There is no information regarding whether any reimbursement/ incentives were given to the participants (E.g. For transport, refreshments etc.) and also, I request the authors the mention a word on how data saturation was reached in the interviews and discussion.

Though it was mentioned in the study’s protocol, it is worth mentioning who were the study participants for 8 FGDs in the methods section (Men and women with under-five children and older family members fathers, mothers and their family members) for easy understanding of the readers.

The authors should describe the distribution of participants in the FGD, whether any group (mothers/ grandparents/fathers) dominated the discussion. It is important to mention this in the methods section as it is susceptible to bias. The group and individual opinions can be swayed by dominant participants. Also, it is good to mention how this was taken care during the FGDs.

Results

The data appear to be sound and the results section elaborately presents the output of the processes described in the methods section.

Line 176- There is a discrepancy in the number of participants for FGDs. In methods section line 131, says 8-11 participants. This should be corrected.

Table 1- Point 1.4, Literacy of father, the 3rd column does not add up to 46. The sub-heading age in years and median (IQR) and n (%) should be shifted to 3rd column for easy understanding.

Line 172- I would suggest the authors to mention the cause of death as per the hospital record for all the 25 child deaths, to understand the leading cause of death among the children of recruited family.

Line 189- It would be good to show the coding tree to describe how the codes that were generated from the data were translated into categories and then to themes.

Line 284- “Mothers were supported by husbands and family members”. Does it include in-laws as well? I would suggest the authors to mention whether any support was given to the mothers from their in-laws during the grief period.

Table 2- It is better to use test of significance to find whether there is statistically significant difference exists among the groups.

The discussion and conclusion are well balanced and adequately supported by the data.

Lines 510 and 527- should be specific and not generic. I suggest the authors to write specific recommendations at the family level, community level and health system level and what should be done to address various factors as mentioned in the results section associated with grief, by whom and at what level of healthcare system this should be implemented.

6. PLOS authors have the option to publish the peer review history of their article (what does this mean?). If published, this will include your full peer review and any attached files.

Reviewer #1: No

---

## [Author Response · Author response to Decision Letter 0]

10 Dec 2020

Manuscript Ref no: PONE-D-20-30229

Title: Grief reaction and psychosocial impacts of child death and stillbirth on bereaved North Indian parents

Journal: PLOS ONE

Response to Editorial comments

The PLOS ONE style templates can be found at

Response: We have made the necessary revisions in the manuscript style and file naming, as per the guideline. 

Response: We have included the title page in the main document as advised. 

"This study is funded to The INCLEN Trust International by Bill and Melinda Gates

Foundation (OPP1184205) through Indian Council of Medical Research (no 5/7/1504/2016-

CH). The funders had no role in study planning, conduct, analysis and manuscript preparation."

"None"

Response: We have removed the funding statement from the main document. (Page 33, Lines 599-601)

The funding statement to be updated as follows. We request to update the funding statement online. 

"This study is funded to The INCLEN Trust International by Bill and Melinda Gates

Foundation (OPP1184205) through Indian Council of Medical Research (no 5/7/1504/2016-

CH). The funders had no role in study planning, conduct, analysis and manuscript preparation."

"None"

Response: Please update the authors Competing Interest statement to 

"The authors have declared that no competing interests exist.”

Response: We are submitting the transcripts of IDIs and FGDs as supplementary file (S2 File).

We are submitting the perinatal grief scale scores for the participants as supplementary file (S4 File).

Response: We have updated the Ethical considerations section in the main text (page 12, lines 185-190) and deleted the ethics statement from Acknowledgement/Declaration section (Page 33, Lines 586-590). 

Response: We have added the section on Supplementary files/documents with the titles (Page 41). We have also mentioned the Supplementary files/documents in the main text as appropriate. 

Reviewers' comments:

Reviewer #1: This is a well-written qualitative paper which aims to study the social, emotional, and psychological impact of child death and stillbirths on parents and their families along with the coping strategies in North Indian context. I appreciate the authors for their commendable work as this is particularly important and less investigated topic of women and child health.

Response: Thank you very much for the encouraging words. 

Comments

Title

I would suggest the authors to add “a qualitative study” at the end of the title as the readers will know the type of study was conducted for this research question.

Response: As suggested, we have revised the title and added “a qualitative study”. 

The revised title now reads as “Grief reaction and psychosocial impacts of child death and stillbirth on bereaved North Indian parents: a qualitative study” (Title, Page 1 and Page 5)

Abstract

The abstract summarizes the rationale for the study and the methods used for this study, as well as the key findings.

Response: Thank you. No action needed. 

Introduction

The introduction includes important information about the global burden of neonatal and infant mortality, the bereavement and grief caused to the mother and family after infant death and shows the paucity of information on the social and psychological impact after child death in North India.

Line 54- I would suggest the authors to write few more lines on the epidemiology of child death in India.

Response: As suggested, we have added few lines on the epidemiology of child death and stillbirths in India. (Introduction, page 7, lines 54-61)

At the end of line 54, authors should discuss about the consequences of child death in the family and then the statements on bereavement and grief should be discussed, as it would give the readers a sense of continuity. Right now, it looks separate because the burden was discussed in the initial few lines of Introduction followed by definition of bereavement and grief.

Response: We have added a sentence after the mortality epidemiology to link between the two segments. The bereavement and grief follows the sentence. 

The sentence added is “Death of a child is one of the most severe, shattering and overwhelmingly painful event for parents.” (Introduction, Page 7, line 63-64)

Line 85- Authors had mentioned “causes of a large proportion of child, infant deaths and stillbirths remain unknown”. This statement should be removed as the leading causes of infant and neonatal deaths are well-documented. I would suggest the authors to write a few lines on the common causes of neonatal and infant deaths in India along with the description of epidemiology of child deaths in the 1st paragraph of Introduction.

Response: As suggested by reviewer, we have removed the statement and added the epidemiology in 1st paragraph. (Introduction, page 7, lines 54-61)

Methods

The methods section provide details on how the participants were chosen, how the interview was conducted, and all the required information needed to understand the whole process of the study.

Line 114- I would suggest the authors to provide some information on how many refused to participate in the study among those who were contacted to find whether there is any systematic difference in the socio-demographic characteristics between the participants and non-participants.

Response: In total during the reference period, there were 45 child deaths, 52 neonatal deaths and 60 stillbirths. Out of these 69 parents were not traceable due to incomplete address or no contact number. We were able to contact 30 eligible parents with child death, 28 parents with neonatal death and 40 parents with stillbirth. Out of these approached, 13 parents with child death, 12 parents with neonatal death and 22 parents with stillbirth consented for IDI. We have added the same to text also. (Methods, page 11, lines 129-134)

Line 143- There is no information on the internal validity (Cronbach’s alpha) of the scale used in this study.

Response: The Cronbach’s alpha estimated was 0.91 for the perinatal grief scale used among the participants. This has been added to the text. (Methods, Page 13, lines 194)

I suggest the authors to comment on the fidelity of the transcription. Whether it was read out to participants after each session to check for correctness or any other method was adopted.

Response: The questions were readout to the participants and the responses were recorded. Any response needed clarification were clarified or needed further exploration were explored or probed. These have been added to the text. (Page 12, lines 169-172)

While transcription, the audio recordings were used. The transcripts were checked by another research team member for correctness. This has been mentioned in the text. (Page 11, lines 178-179)

There is no information regarding whether any reimbursement/ incentives were given to the participants (E.g. For transport, refreshments etc.) and also, I request the authors the mention a word on how data saturation was reached in the interviews and discussion.

Response: No payment or reimbursement/incentive was made to any participant. The statement has been added in the text. (Page 12, Lines 172) 

Regarding data saturation, we have added the statement in Methods section. (Methods, Page 10, Lines 150-152). 

Though it was mentioned in the study’s protocol, it is worth mentioning who were the study participants for 8 FGDs in the methods section (Men and women with under-five children and older family members fathers, mothers and their family members) for easy understanding of the readers.

Response: We have added the participants for FGDs in the methods as suggested. (Page 11, Lines 135-137)

The authors should describe the distribution of participants in the FGD, whether any group (mothers/ grandparents/fathers) dominated the discussion. It is important to mention this in the methods section as it is susceptible to bias. The group and individual opinions can be swayed by dominant participants. Also, it is good to mention how this was taken care during the FGDs.

Response: The facilitator tried to moderate the discussion among participants and encouraged all participants to contribute. The quality of discussion was documented by sociogram. We have added this to the text in Methods section. (Page 12, Lines 158-159). The sociograms are attached as supplementary document (S3 File). 

Results

The data appear to be sound and the results section elaborately presents the output of the processes described in the methods section.

Line 176- There is a discrepancy in the number of participants for FGDs. In methods section line 131, says 8-11 participants. This should be corrected.

Response: The distribution of participants for FGDs were: 

• Fathers: Total 18 (FGD-1, n=9, FGD-2, n=9)

• Mothers: Total 19 (FGD-1, n=11, FGD-2, n=8)

• Grandfathers: Total 17 (FGD-1, n=9, FGD-2, n=8)

• Grandmothers: Total 18 (FGD-1, n=9, FGD-2, n=98)

Thus we have written the participants as 8-11 in the methods section. 

Table 1- Point 1.4, Literacy of father, the 3rd column does not add up to 46. The sub-heading age in years and median (IQR) and n (%) should be shifted to 3rd column for easy understanding.

Response: We apologise for the typo error. The number for category <5th standard is 6. It has been corrected. 

The correction in the table has been done as suggested. (Table 1, Page 14-16)

Line 172- I would suggest the authors to mention the cause of death as per the hospital record for all the 25 child deaths, to understand the leading cause of death among the children of recruited family.

Response: The causes of death for the 25 children (12 neonates and 13 post-neonatal children) have been added as supplementary document (S5 file). The same has been added to the text in Results section. (Page 14, Lines 212-213).

Line 189- It would be good to show the coding tree to describe how the codes that were generated from the data were translated into categories and then to themes.

Response: The coding tree of the themes and categories are given as supplementary document (S6 File). The same has been added to the text in Results section. (Page 16, Lines 230-231).

Line 284- “Mothers were supported by husbands and family members”. Does it include in-laws as well? I would suggest the authors to mention whether any support was given to the mothers from their in-laws during the grief period.

Response: The mothers were supported by in-laws and also their mothers and sisters. The same has been added to the text as suggested. (Page 20, Lines 323-324)

Table 2- It is better to use test of significance to find whether there is statistically significant difference exists among the groups.

Response: The mothers with stillbirth had higher PGS scores for total and subscales than the mothers with child/neonatal death. Statistical significant differences were observed in the total PGS score (p=0.02) and one sub-scale (despair, p=0.02) between the mothers with stillbirth and mothers with child/neonatal death. It has been added to the text in Results. (Page 23, Lines 403-405) 

The discussion and conclusion are well balanced and adequately supported by the data.

Lines 510 and 527- should be specific and not generic. I suggest the authors to write specific recommendations at the family level, community level and health system level and what should be done to address various factors as mentioned in the results section associated with grief, by whom and at what level of healthcare system this should be implemented.

Response: We thank reviewer for the suggestion. We have added the recommendations for different levels, as advised. We hope that these are appropriate. (Page 30, Lines 553-561)

---

## [Editor Report · Decision Letter 1]

11 Dec 2020

Grief reaction and psychosocial impacts of child death and stillbirth on bereaved North Indian parents

PONE-D-20-30229R1

Dear Dr. Das,

We’re pleased to inform you that your manuscript has been judged scientifically suitable for publication and will be formally accepted for publication once it meets all outstanding technical requirements.

Kind regards,

Vijayaprasad Gopichandran

Academic Editor

PLOS ONE
---

## [Editor Report · Acceptance letter]

15 Jan 2021

PONE-D-20-30229R1 

Grief reaction and psychosocial impacts of child death and stillbirth on bereaved North Indian parents: a qualitative study   

Dear Dr. Das:

I'm pleased to inform you that your manuscript has been deemed suitable for publication in PLOS ONE. Congratulations! Your manuscript is now with our production department. 

Kind regards, 

on behalf of

Dr. Vijayaprasad Gopichandran 

Academic Editor

PLOS ONE